# Electrospun Polycaprolactone/Chitosan Nanofibers Containing *Cordia myxa* Fruit Extract as Potential Biocompatible Antibacterial Wound Dressings

**DOI:** 10.3390/molecules28062501

**Published:** 2023-03-09

**Authors:** Amal A. Alyamani, Mastafa H. Al-Musawi, Salim Albukhaty, Ghassan M. Sulaiman, Kadhim M. Ibrahim, Elsadig M. Ahmed, Majid S. Jabir, Hassan Al-Karagoly, Abed Alsalam Aljahmany, Mustafa K. A. Mohammed

**Affiliations:** 1Department of Biotechnology, College of Science, Taif University, P.O. Box 11099, Taif 21944, Saudi Arabia; 2Department of Clinical Laboratory Science, College of Pharmacy, University of Al-Mustansiriyah, Baghdad 14022, Iraq; 3Department of Chemistry, College of Science, University of Misan, Amarah 62001, Iraq; 4College of Medicine, University of Warith Al-Anbiyaa, Karbala 56001, Iraq; 5Division of Biotechnology, Department of Applied Sciences, University of Technology, Baghdad 10066, Iraq; 6Department of Plant Biotechnology, College of Biotechnology, Al-Nahrain University, Baghdad 64074, Iraq; 7Department of Medical Laboratories, College of Applied Medical Sciences, University of Bisha, P.O. Box 551, Bisha 61922, Saudi Arabia; 8Department of Clinical Chemistry, Faculty of Medical Laboratory Sciences, University of El Imam El Mahdi, P.O. Box 27711, Kosti 209, Sudan; 9Department of Internal and Preventive Medicine, College of Veterinary Medicine, University of Al-Qadisiyah, Al-Diwaniyah 58002, Iraq; 10Department of Medical Basic Sciences, College of Applied Medical Sciences, University of Bisha, P.O. Box 551, Bisha 61922, Saudi Arabia; 11Radiological Techniques Department, Al-Mustaqbal University College, Hillah 51001, Iraq

**Keywords:** *Cordia myxa*, polycaprolactone, chitosan, scaffolds, electrospun

## Abstract

The goal of the current work was to create an antibacterial agent by using polycaprolactone/chitosan (PCL/CH) nanofibers loaded with *Cordia myxa* fruit extract (CMFE) as an antimicrobial agent for wound dressing. Several characteristics, including morphological, physicomechanical, and mechanical characteristics, surface wettability, antibacterial activity, cell viability, and in vitro drug release, were investigated. The inclusion of CMFE in PCL/CH led to increased swelling capability and maximum weight loss. The SEM images of the PCL/CH/CMFE mat showed a uniform topology free of beads and an average fiber diameter of 195.378 nm. Excellent antimicrobial activity was shown towards *Escherichia coli* (31.34 ± 0.42 mm), *Salmonella enterica* (30.27 ± 0.57 mm), *Staphylococcus aureus* (21.31 ± 0.17 mm), *Bacillus subtilis* (27.53 ± 1.53 mm), and *Pseudomonas aeruginosa* (22.17 ± 0.12 mm) based on the inhibition zone assay. The sample containing 5 wt% CMFE had a lower water contact angle (47 ± 3.7°), high porosity, and high swelling compared to the neat mat. The release of the 5% CMFE-loaded mat was proven to be based on anomalous non-Fickian diffusion using the Korsmeyer–Peppas model. Compared to the pure PCL membrane, the PCL-CH/CMFE membrane exhibited suitable cytocompatibility on L929 cells. In conclusion, the fabricated antimicrobial nanofibrous films demonstrated high bioavailability, with suitable properties that can be used in wound dressings.

## 1. Introduction

The skin acts as a physical barrier to keep germs out of possible infection sites in living beings. However, traumas such as fractures, heat burns, lacerations, or surgical incisions can damage the skin’s structure and function and occasionally cause the skin to lose its protective function due to a disruption in the tissue’s healthy relationship caused by wounds [1,2]. Local antibacterial and dressing therapies are widely utilized to reduce drug resistance in the case of treatment with systemic antibiotics [3]. Additionally, wound healing is a complex process that includes epithelialization, matrix deposition, cell proliferation, tissue regeneration, and inflammation [4]. As a result, the optimal dressing should be moisturizing and breathable, absorb exudate, have anti-inflammatory, antibacterial, and antioxidative properties, and encourage cell proliferation and wound healing [5,6,7]. Due to their adjustable nanostructures and functionalities, nanomaterials, including the bactericidal performance of nanoparticles (NPs) and nanofibers, have recently received a significant promotion for the treatment of wounds [8,9,10,11,12]. Nanofibers are among the nanomaterials mentioned above that have high specific surface areas, microscopic pore diameters, moisturizing qualities, the capacity to allow air exchange, and morphologies that resemble the extracellular matrix; these features make nanofibers attractive options for wound dressings [13,14].

A remarkable method for producing nanofibers on a large scale from a polymer suspension is electrospinning. Under the influence of an electric field, electrospun fibers may be spun to nanoscale sizes [15,16]. Numerous natural and synthetic polymers have been electrospun into fibers, which have tremendous potential for medication delivery for the treatment of burn injuries [17]. Due to its appropriate mechanical behavior and biocompatibility, polycaprolactone (PCL), a biodegradable, stable, and non-hazardous synthetic polymer has been widely used in electrospun nanofiber membranes [18,19]. However, due to their hydrophobic properties, monomer PCL nanofibers may not be the best substrate for promoting cell growth and migration [20]. To enhance the surface-wetting capability of nanofibers, varieties of hydrophilic materials have been combined with PCL. Chitosan (CH) is a natural polysaccharide with biodegradable, biocompatible, and non-toxic properties and is thus proposed to be a safer substrate for use in biomedical applications, such as tissue engineering, the delivery of pharmaceutical drugs, and as wound dressings [21,22,23]. Chitosan has been studied extensively as a wound dressing because it can serve as a promoter of proliferation, an antibacterial agent, and an activator of macrophages [24]. Moreover, the inclusion of CH in PCL nanofibers may enhance the surface hydrophilicity of scaffolds [25].

The *Cordia myxa* tree, also known as “Bamber” in Iraq, was discovered to contain highly bioactive compounds. *Cordia myxa* can be used to treat respiratory tract infections as an expectorant and demulcent and can also act as a diuretic and anti-diarrheal medication. The compounds isolated from the genus *Cordia* have been identified as having antiviral, tumor cell cytotoxic, anti-inflammatory, and free radical scavenging properties [26,27,28,29]. Here, we propose that *Cordia myxa* fruit, by its antioxidant and antimicrobial properties, could eliminate chemicals such as reactive oxygen species generated via inflammatory responses and minimize the slow healing of wounds caused by infection. Herein, *Cordia myxa* fruit extract was loaded into poly(-caprolactone) chitosan-based nanofibrous mats to improve antimicrobial efficacy and biocompatibility for accelerated wound healing. SEM, XRD, and FT-IR studies of the morphological and structural properties of the fabricated nanofibers and their surface wettability, water absorption, and weight loss were also characterized. MTT assays on the L929 cell line were performed to assess the cytotoxicity. In addition, antibacterial properties against *Escherichia coli*, *Salmonella enterica*, *Staphylococcus aureus*, *Bacillus subtilis*, and *Pseudomonas aeruginosa* were studied via the inhibition zone test, which implied that the obtained PCL-CH/CMFE is a promising wound dressing for skin injuries.

## 2. Results and Discussion

The current study focused on CMF fruit extracts stabilized on polycaprolactone/chitosan nanofibers for medical uses, such as antimicrobial properties for wound dressing applications. The rationale behind immobilizing plant materials with pharmacological significance, particularly for healing wounds, on a polymer matrix is fascinating [30]. The polymer matrix can encourage properties such as chemical and antimicrobial effectiveness to the plant extracts owing to its small size and surface area-to-volume ratio. Electrospinning was utilized to prepare PCL/CH nanofibrous scaffolds in the current study, where the CMF extract was immobilized into the created nanofibers. In spite of the fact that medicinal CMF extract has notable wound healing and antibacterial actions, it is preferable to improve them via immobilizing the extract on the nanofibrous surface of a scaffolding [31]. Electrospinning polymer nanofibers of PCL/CH containing a varied quantity of CMF extract was performed according to the criteria outlined in the methods section. Table 1 summarizes the composites and electrospinning parameters that were employed to create the samples.

Figure 1 reveals SEM micrographs of PCL/CH electrospun nanofibers and CMF fruit extract-loaded PCL/CH nanofibers that showed a nano-scaled fibrous architecture without the existence of beads. This was obtained by utilizing the optimal spinning conditions used in this research. Round-shaped PCL nanofiber scaffolds with a smooth surface are optimum characteristics. Since plant extracts induce clusters to develop at branching points, a uniform fiber surface appears to be essential, even though the fiber diameter of the CMF-PCL/CH nanofiber scaffolds did not change significantly. However, due to the immobilization of the extract, the SEM micrographs of the CMF extract-loaded fibers seem to have a slightly swollen morphology. The diameter size of fibers was calculated using ImageJ, and it was shown to be in the range of 97 ± 7 nm for PCL nanofibers. CH tends to make the electrospinning suspension more polar due to its charged functional groups that cause fibers to stretch in an electrical field [32]. The diameter of PCL/CH fibers was 197 ± 23 nm, while the diameter of CH/PCL/CMF fibers was 295 ± 78 nm, with a slight increase due to swelling.

To demonstrate the crystalline structure of CMFE-loaded PCL/CH nanofibers, X-ray diffraction (XRD) was investigated. XRD showed that the electrospun nanofibers contained crystals (Figure 2). The solvent medium, parameters, applied voltage, and polymer characteristics (molecular weight) all affect the crystal structure [33]. XRD was taken to analyze the modifications in the crystalline structure of the pristine electrospun PCL/CH/CMF nanofiber membrane. This is demonstrated in Figure 2. Two diffraction peaks at transmittance angles of 21.4° and 23.8° were observed in both cases, which were related to the semi-crystalline behavior of PCL [34,35,36].

The FTIR test was employed to investigate the functional groups found in PCL, CH, CMFE, and PCL/CH/CMFE (Figure 3). The pure PCL membrane had characteristic peaks at 2938, 2865, 1730,1408, 1106, and 730 cm^−1^, which are related to the (CH_2_), (CH_2_), (C=O), (CH_2_), (O-C), and (CH_2_) vibrations [37]. The FTIR spectra of chitosan showed peaks at 3440 cm^−1^, 1636–1650 cm^−1^, and 1107 cm^−1^ that contribute to chitosan groups, as stated in the FTIR spectra of chitosan [38]. The broad peak in the spectra of pure CMFE shown in Figure 3 at 3404 cm^−1^ revealed the existence of both free hydroxyl groups and hydroxyl groups involved in hydrogen bonding. The sharp peak at 1712 cm^−1^ may be caused by the carbonyl group (C=O), which could also point to the existence of compounds containing carboxylic acid. Similar results suggest that the *C. dichotoma* fruit extract contains some uronic acid in previous studies [39]. The band at 1434 cm^−1^ is caused by the C=C stretching vibration in aromatic rings, such as those present in phenolic and flavonoid compounds.

The spectrum of the PCL/CH/CMFE mat (Figure 3) showed no major extra peaks or deviations from the peaks of the pure materials. This confirmed that CFE remained mostly intact and unreacted within the nanofibers. The PCL/CH/CMFE spectrum, as shown in Figure 3, suggests that there were no significant extra peaks or changes from the peaks of the pure materials. This confirmed that CMFE was mostly unreacted and retained inside the fibers.

Stress–strain profiles exhibit two nanofiber samples made from pure PCL 12 *w*/*v* and PCL/CH/CMF 5% volume ratios of 70/30 (Figure 4A and Table 2). Adding the natural polymer to the scaffold reduces its mechanical strength. Lower fiber strength is detected in the sample containing 12% PC, in addition to enhanced flexibility and elongation. PCL/CH/CMFE 3% membranes presented a stress (MPa) of 20.1107 in a dry state, whereas PCL demonstrated 24.8931 MPa under the same conditions (Figure 4B).

The mechanical behavior of wet membrane scaffolds was examined after being immersed in phosphate-buffered saline (PBS) for 24 h, and standard stress–strain curves were produced utilizing standard tensile stress–strain testing (Figure 4). Table 2 demonstrates that the PCL’s extension, thickness, tensile modulus, and stress were reduced by the addition of CH and CMFE. The PCL has 69.2% porosity, while the PCL/CH/CMFE has 76.9%.

Wettability is an essential factor to consider when choosing a wound dressing since it influences cell adherence, proliferation, and the ability to absorb exudates. The water contact angle can be used to determine the wettability of a surface. The water contact angle was measured to determine the behavior of the composite PCL/CH/CMFE mats and assess the hydrophilicity alterations in nanocomposite scaffolds. As presented in Figure 5, results obtained for PCL-12% film exhibit poor hydrophilicity with an average contact angle of 125.5°, which is in line with the hydrophobic nature of the polymer. The contact angle value of PCL decreased to 99.4°, 73°, and 47.4° after the addition of CH at 2%, CMFE at 3%, and CH at 2% + CMFE at 3%, respectively.

It has been demonstrated that greatly swelling matrices promote cell growth, adhesion, and internal migration of scaffolds [39]. Anionic and cationic polyelectrolytes improve the electrical conductivity of the electrospinning solution [40].

The swelling’s capacity to absorb wound exudates keeps the region surrounding the wound dry and reduces the risk of infection. The swelling ratio was increased by the use of CMFE in this study. The swelling behavior of the PCL/CH/CMFE nanofibrous performed better, with values of 101.2 ± 6.1 as compared to the PCL/CH mat, which was 86.7 ± 7.7, as shown in Table 2. The results of the wettability test supported our swelling findings. The incorporation of CH and CMFE within the PCL matrix may present some hydrophilic groups, such as NH and OH, on the surface of the nanocomposite membranes. Therefore, the wettability of the PCL scaffold can be seriously modified by the addition of CH/CMFE nanofiber. The higher hydrophilicity of the resultant scaffold is thought to be due to hydrophilic groups, such as hydroxyl and carboxyl groups, which are present in the CMF structure as identified by the GC-MS test. These findings are in agreement with the result of Allafchian et al., who reported that the contact angle for PCL is 120°, while after the addition of quince seed mucilage, the contact angle value of PCL decreased to about 40° [41].

The in vitro release of CMFE from the PCL/CH fibrous mat is depicted in Figure 6. The release curve showed that larger CMFE concentrations were released at equivalent time points. The PCL/CMFE 3% mats released 94.5% within the first 24 h. Thereafter, the slope of the curves decreased, indicating a more gradual release, which extended to 120 h. The maximum CMFE concentrations after 120 h were 62.1 ± 2.2 μg/mL for the PCL/CMFE 3% mat.

The antibacterial activity of the nanofibrous mat is important for the healing of skin lesions. Therefore, the antibacterial effectiveness of the produced CMFE-loaded PCL/CH nanofibrous mats was examined in this section against the most prevalent bacterial species that cause wound infections, including *E. coli*, *S. Enterica*, *P. aeruginosa*, *S. aureus*, and *B. subtilis*. Figure 7 illustrates a PCL/CH scaffold with and without CMFE antibacterial activity. PCL/CH fibers without CMFE extract exhibit antibacterial activity with various inhibition zones against *B. subtilis* (11.47 mm), *S. aureus* (12.5 mm), and *E. coli*, *S. enterica*, and *P. aeruginosa* with 22.5 mm, 22.1 mm, and 15.4 mm, respectively. PCL/CH/CMFE demonstrated significant antibacterial activity against *S. aureus* (21.31 ± 0.17 mm with IZ at 100 mg/mL concentration), *E. coli* (31.34 ± 0.42 mm), and *S. enterica*, *B. subtilis*, and *P. aeruginosa* (30.27 ± 0.57, 27.53 ± 1.53, and 22.17 ± 0.12 mm, respectively).

As Gram-positive bacteria, *S. aureus* and *B. subtilis* are believed to have thicker cell walls and are, therefore, more resistant to action by chitosan than *E. coli* [42]. Additionally, it has been shown that chitosan has a much stronger affinity toward Gram-negative bacteria in comparison to Gram-positive bacteria [43]. It is well known and has been thoroughly researched that chitosan has antibacterial properties that are connected to its cationic structure, which results from its protonated amine groups when it comes into contact with liquids [44]. It is possible that this is connected to the positively charged NH_2_ groups of CH, which absorb and desorb negative and positively charged Gram-negative and Gram-positive bacterium, respectively, due to electrostatic forces [45]. It is important to emphasize that CMFE extracts added antibacterial action to chitosan-based nanofibrous mats against Gram-negative bacteria in addition to providing antibacterial activity against Gram-positive bacteria. This is attributed to CMFE’s natural ability to *combat E. coli*, *S. enterica*, *P. aeruginosa*, *S. aureus*, and *B. subtilis* [28,46] and its synergistic antimicrobial effect with chitosan against the above pathogens. Thus, the synergistic antibacterial action of PCL/CH-based nanofiber and coupled CMFE exhibited encouraging results to enhance antibacterial wound dressings. Further research is necessary to fully comprehend the mechanism of growth suppression in *B. subtilis*, *S. aureus*, *E. coli*, and *S. enterica*, as well as the detailed mode of action of the bioactive components of CMFE.

This study showed positive results concerning the antimicrobial activity caused by PCL/CH/CMF to *S. aureus*, *E. coli*, *S. enterica*, *B. subtilis*, and *P. aeruginosa*. This corresponds to the previous study conducted by Stasiuk et al., [47] who reported antibacterial activities against *S. aureus* and *E. coli*. These results suggest that the ethanol extract of CMF can be utilized to inhibit foodborne diseases caused by microbial pathogens such as *S. aureus*, *E. coli*, *S. enterica*, *B. subtilis*, and *P. aeruginosa*. The current results are consistent with the previously published findings of Hamdia et al. [48] on the antibacterial activity of Cordia fruit extracts. These authors showed that *C. myxa* extracts (aqueous and alcoholic) exhibited concentration-dependent inhibition zones against *P. fluorescens*, *S. enterica*, *S. dysenteriae*, and *E. coli*. The results suggest that fruit extract could be used to reduce microbial infections caused by *S. aureus*, *E. coli*, *S. enterica*, *B. subtilis*, *P. aeruginosa*, *A. brasiliensis*, and *S. cerevisiae*.

To evaluate the cytotoxicity of the nanofibers, we performed the MTT test. The viability of cells cultured in various nanofiber suspensions varied, as seen in Figure 8. Compared to the pure PCL membrane, the PCL-CH/CMF membrane exhibited almost negligible cytotoxicity. The enhanced biocompatibility of CMF may have increased cell viability. Results showed that dermal fibroblast cells were not significantly cytotoxic to the nanofiber scaffolds of PCL/CH/CMFE. The PCL/CH/CMFE electrospun nanofibers have improved surface area, density, and pore size characteristics. This occurred by changing critical manufacturing parameters, such as the distance between the needle and collector, electrical voltage, and the volume of the pumped polymer. Nanofibers have grown in importance in biotechnology due to their prospective uses and huge surface area, which might be employed to make a good matrix for biological activity [49]. It is crucial to employ biocompatible chemicals such as CH and CMFE; because of their greater bioactivity compared to synthetic polymers, natural polymers such as CH have demonstrated promising outcomes in vitro and in vivo [50]. Reactive oxygen species (ROS) depletion-based therapeutics like CMFE may help to reduce inflammation and facilitate a smooth transition from the inflammatory to the proliferative phase of the cell cycle. According to earlier research, natural antioxidants such as plant flavonoids may prevent free radical chain reactions or scavenge ROS to reduce the oxidation of cellular components and promote cell development [51]. The specific mechanisms by which CMFE stimulates the production of fibroblast growth factor and cell proliferation are still unknown. According to earlier studies, CMFE interacts with the growth factor receptors on fibroblasts to promote cell activity and proliferation. One of the key hypothesized mechanisms of topical CMFE activity on the wound area was related to the plant extract’s chemotactic action, which may have attracted inflammatory cells in addition to having antimicrobial properties [52].

## 3. Materials and Methods

### 3.1. Chemicals

Polycaprolactone (PCL, Mn 80,000 g/mol), chitosan (CH) (≥85% (degree of deacetylation) Mw 100 kDa, acetic acid, and formic acid were purchased from Sigma Aldrich at concentrations of 0.2 M and 0.6 M, respectively. All the compounds employed were of analytical reagent quality and required no further processing.

### 3.2. Plant Collection and Extraction

Fresh *C. myxa* fruits (Figure 9) were harvested from trees in Baghdad, Iraq, between July and August 2020, and botanists from Al-Nahrain University confirmed their authenticity. The harvested plant material was packaged in a polyethylene bag to avoid moisture loss during transport to Al-Nahrain University’s Research Center laboratory in Baghdad.

The procedure for *C. myxa* fruit extract was conducted according to the previously published procedure reported by El-Massry et al. [53]. Briefly, the fruits were cleaned and thoroughly inspected to remove those that were physically or microbiologically damaged. The gathered fruits were cleaned with tap water and passed through a deionized water solution. The harvest was collected, dried in the shade, and then processed into a powder. Then, 100 g of powder was mixed with 250 mL of ethanol in a round-bottom flask utilizing a magnetic stirrer for three hours at 25 °C, and the mixture was filtered under vacuum (using Whatman No. 1 filter paper, Cytiva, NY, USA). The materials of the filter paper were transferred to the conical flask once again, and the procedure was repeated. The extract was pooled and dried using a rotating evaporator. For yield computation, the resultant crude extract was weighed and kept at 4 °C until required. As a solvent for extraction, a 70% ethanolic solution was utilized. In the following studies, the fruit sample was diluted in ethanol at an optimal concentration. The residual material was preserved at −20 °C until further use. All the tests were carried out within 72 h of extraction.

### 3.3. Electrospun

The electrospinning variables, PCL and CH concentration, were assessed with minimal modifications based on a recently published study [32]. For the electrospun method, the optimum voltage, feeding rate, and distance between the needle and the collector were set at 12 kV, 0.6 mL/h, and 21 cm, respectively, and the fibers were collected using an aluminum sheet. A suspension of 12% PCL, 2% CH, and 5% CMF was prepared separately using a 30/70 solvent mixture of acetic acid and formic acid. Then, a mixture of PCL/CH/CMF in the ratio of 3/1/2 was produced, and it was placed on a magnetic stirrer for 15 h to create a uniform mixture. As-prepared suspensions were placed into a 5 mL plastic syringe, whose nozzle was an 18-gauge stainless steel tip. For the gathering of the nanofibrous mats, several voltages and flow rates were applied to each of the blend suspensions, while the optimal values that produced uniform nanofibers were chosen. A total of 20 cm was maintained between the tip and the collection. Materials were gathered on a ground collector coated in aluminum plates and cotton gauze. During collection, a fixed collecting surface was utilized, while the spinneret moved transversely at 100 mm/s and was 100 mm in diameter.

### 3.4. Morphological Features

SEM (TESCAN, MRA-3, Brno, Czech Republic) was used to analyze the morphological properties of the CMF/PCL/chitosan scaffold nanofibers, which were coated with an ultrathin 5-nm gold undercurrent and a voltage of 6 kV. XRD at the Cu_Kα_ wavelength (0.15405 nm) was used to investigate the crystalline phase of CMFE-loaded PCL/CH. The current utilized was 25 mA, while the voltage used was 40 kV.

### 3.5. FTIR

FTIR (IR Tracer 100, Shimadzu, Japan) measurements in the 500–4000 cm^−1^ range were used to investigate the functional groups in the nanofiber electrospun samples.

### 3.6. Porosity

The tensile strength and elasticity of PCL/CH/CMFE were calculated during the examination of scaffold strength. The porosity of the PCL/CH/CMFE composite was determined using the gravimetric approach as in previously published work [25]:$$p=1-\frac{Ad\;}{Bd\;}\times 100\%$$
where *p* indicates porosity, *Ad* represents the apparent density fibrous mat, and *Bd* indicates the bulk density fibrous mat.

### 3.7. Water Contact Angle

The contact angle was dynamically calculated using the Wilhelmy plate approach [54]. The hydrophilicity or hydrophobicity of the sample was tested and evaluated using water with a surface tension of 72 dyn/cm. The liquid in the container increased until the metal plate’s intended surface was completely submerged in water, at which point it began to descend. The formula below can be used to determine the contact angle.
$$Τcos\;\theta =\frac{Ws}{C}$$
where *T* is the water’s surface tension, *θ* indicates the water contact angle, *Ws* indicates the weight shift, and *C* is the fibrous mat’s diameter.

### 3.8. The Swelling Test

The swelling level of the PCL/CH/CMFE samples was evaluated in PBS solution at a pH of 7.4 for 2 h at room temperature in accordance with the following relationship:$$St=\frac{Ws-Wd}{Wd}\times 100\%$$
where *S_t_* is the swelling test, *Ws* is the weight of each sample after it is immersed in the buffer solution for an hour, and *Wd* is the sample’s initial weight in its dry form.

### 3.9. CMFE Release Profile

A 1 cm^2^ piece of CMFE-loaded PCL/CH mat was soaked in 14 mL of pH 7.4 PBS buffer at 37 °C for various times ranging from 1 to 120 h with moderate stirring. To measure the CMFE concentration, a UV spectrophotometer at 280 nm was used with a calibration curve of CMFE at ranges of 0 to 100 µg/mL, and aliquots of the solution (200 μL) were taken at each time point. To maintain the overall volume constant, a fresh 2 µL of PBS was added to the solution after the same volume was removed.

### 3.10. Antibacterial

*S. aureus*, *E. coli*, *S. enterica*, *B. subtilis*, and *P. aeruginosa* were used for the evaluation of PCL/CH/CMFE in this part. The antibacterial activities of PCL/CH scaffolds with and without CMFE were evaluated by a zone inhibition test. Antibacterial activity was tested against *E. coli*, *S. enterica*, *S. aureus*, *B. subtilis*, and *P. aeruginosa*. Briefly, the synthetic corneal plates were cut to a diameter of 6 mm. UV light was used to sterilize the plates for one hour. Gentamycin was utilized as a control group, and the disks were cultured for 24 h at 37 °C. The growth inhibition zones were then calculated.

### 3.11. MTT Assay

The MTT test was carried out on the normal fibroblast cells, L929 cells, in order to evaluate the biocompatibility effect of the PCL/CH/CMF fibrous mat. Briefly, DMEM enriched with 10% fetal bovine serum, streptomycin (100 g/mL), and penicillin (100 U/mL) were used to cultivate the cells. They were then incubated at 37 °C with 5% CO_2_. Therefore, cells were plated in a multi-well plate with DMEM-soaked fibers at a density of 2.4 × 10^4^ cells per well and incubated for 24, 48, and 72 h. The MTT solution was then incorporated into the media in each well, and the cells were cultivated for an additional 4 h at 37 °C. The final step was dissolving the produced formazan crystals in DMSO and reading the absorbance at 570 nm on a microplate reader.

## 4. Conclusions

In this study, a novel fibrous scaffold formulation was successfully prepared using electrospun PCL/CH fibers loaded with CMFE and was found to be safe for skin L929 fibroblast cells for use in wound dressing applications. After that, their structure, physicomechanical, antimicrobial, and in vitro characteristics were investigated. SEM analysis revealed thin fibers with mean diameters as low as 197 nm and a bead-free shape. PCL/CH fibers loaded with CMFE showed high porosity, promising tensile strength, enhanced hydrophilicity, and biocompatibility. Furthermore, the incorporation of CMFE in the PCL/CH composite was supported by the chemical analysis of the nanofibrous composite by FTIR spectroscopy and structural XRD data. CMFE-loaded PCL/CH fibers were effective against both Gram-positive (*B. subtilis*, *S. aureus*) and Gram-negative (*S. enterica*, *E. coli*, and *P. aeruginosa*) bacteria. The CMFE drug was rapidly released within 24 h. According to the findings, PCL/CH/CMFE electrospun nanofibers demonstrated outstanding promise and had the potential to be employed as wound dressings.

## Figures and Tables

**Figure 1 molecules-28-02501-f001:**
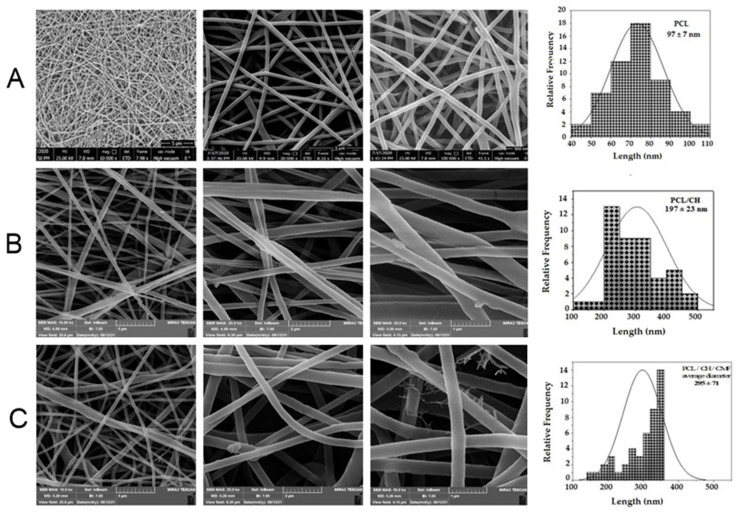
(**A**) SEM images of morphology characterization of different fibrous substrates of PCL (12% *w*/*v*) nanofibers, diameter 97 ± 7 nm, respectively. (**B**) SEM images of PCL (12% *w*/*v*) and CH (2% *w*/*v*) nanofibers, diameter 197 ± 23 nm, respectively. (**C**) SEM images of PCL (12% *w*/*v*) and CH (2% *w*/*v*), CMF (5% *w*/*v*) nanofibers, diameter 295 ± 78 nm, respectively.

**Figure 2 molecules-28-02501-f002:**
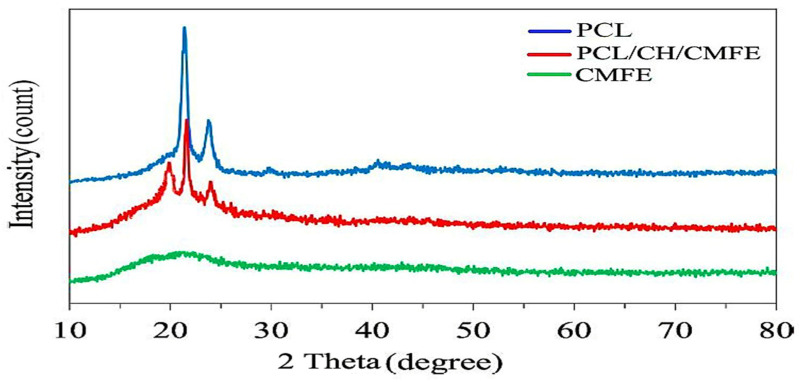
XRD patterns for polycaprolactone (PCL), *Cordia myxa* fruit extract (CMFE), and PCL/CH/CMFE composite nanofibers.

**Figure 3 molecules-28-02501-f003:**
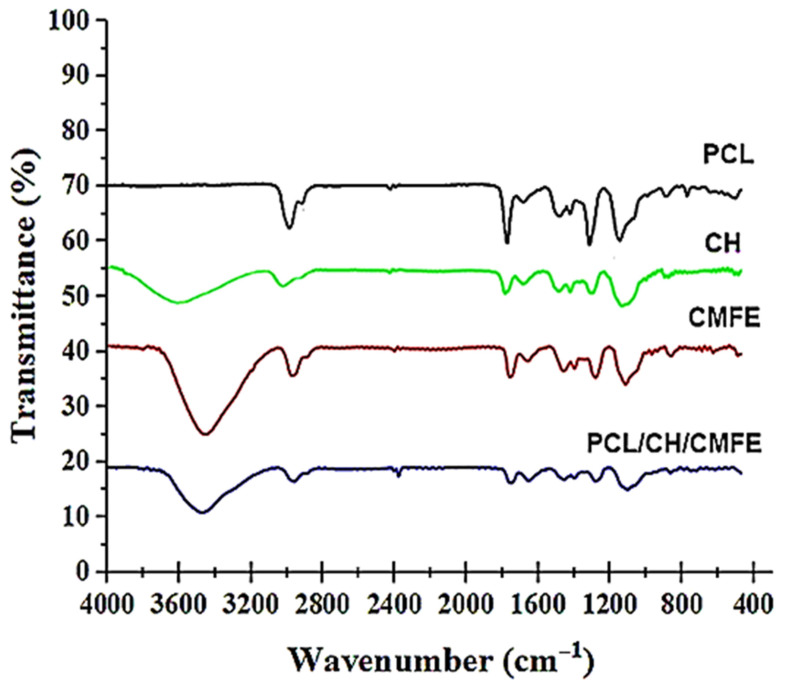
FTIR spectra of polycaprolactone (PCL), chitosan (CH). *Cordia myxa* fruit extract (CMFE), and PCL/CH/CMFE composite nanofibers.

**Figure 4 molecules-28-02501-f004:**
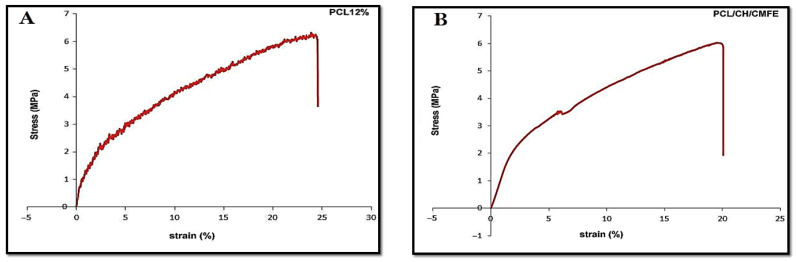
(**A**) Representative stress–strain curve of pure PCL (**B**) PCL/CH containing CMFE.

**Figure 5 molecules-28-02501-f005:**
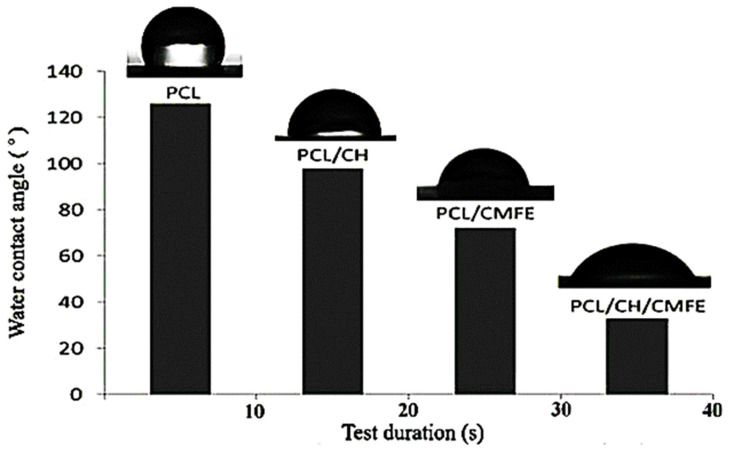
Contact angle characterization of the nanofibrous membranes.

**Figure 6 molecules-28-02501-f006:**
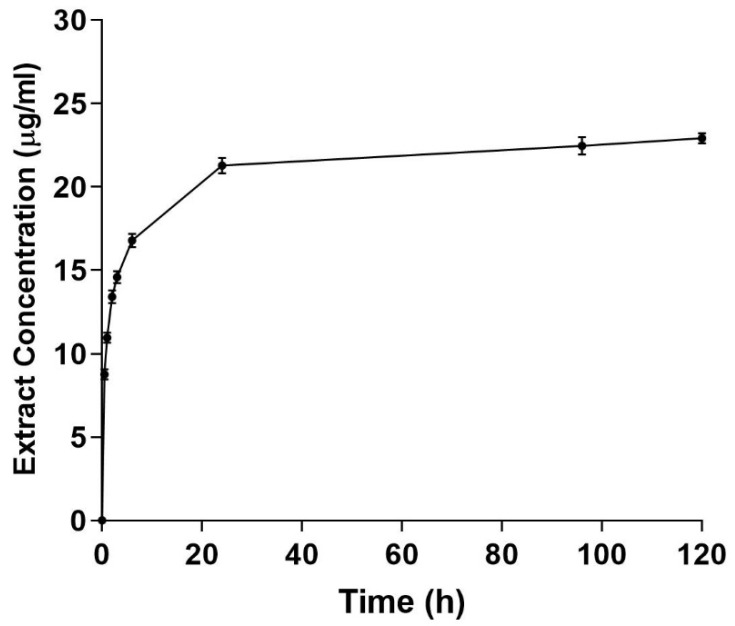
Cumulative extract release from PCL/chitosan 3% mats. Extract concentrations were measured at different intervals in the PBS buffer solution (n = 3).

**Figure 7 molecules-28-02501-f007:**
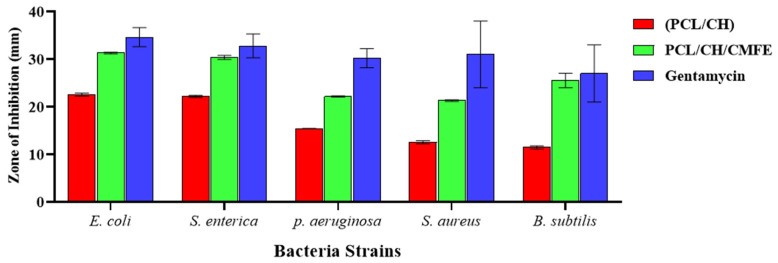
Antibacterial activity of PCL/CH/CMF fibrous scaffold.

**Figure 8 molecules-28-02501-f008:**
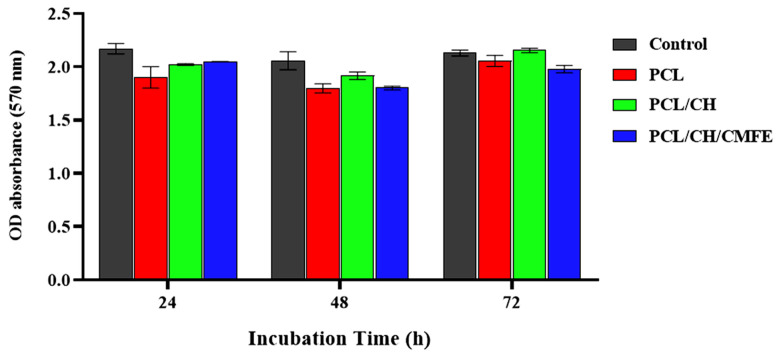
Viability test results (%) for fibroblast cell lines cultured for 24, 48, and 72 h with clear PCL, PCL/CH, and PCL/CH/CMFE mats. The mean and standard deviation are shown as bars (n = 5).

**Figure 9 molecules-28-02501-f009:**
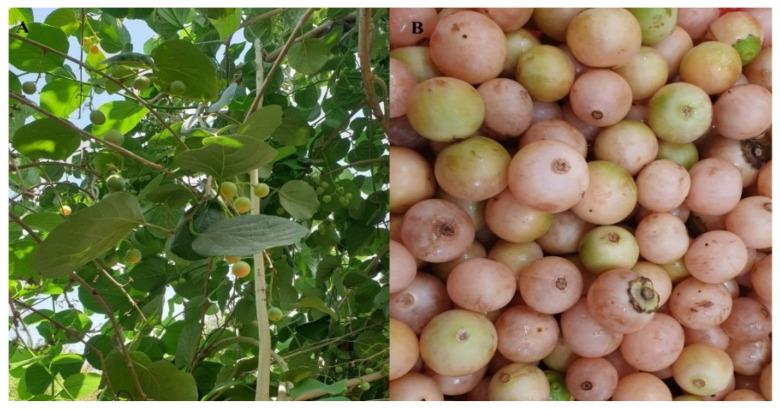
Images represent *Cordia myxa* tree (**A**) and its fruits (**B**).

**Table 1 molecules-28-02501-t001:** The electrospinning settings of fabricated fibrous mats.

Selected Specimen	Conc.	Feed Rate	T.C.D	Volt
PCL	12%	0.5	18	20
PCL/CH	12% + 2%	0.5	20	25
PCL/CH/CMFE	12% + 2% + 5%	0.5	21	20

**Table 2 molecules-28-02501-t002:** Mechanical properties of nanofiber mesh (PCL and PCL/CH/CMFE).

Sample	Extension (mm)	Tensile Modulus (MPa)	Stress (MPa)	Porosity (%)	Thickness
PCL pure	21.5227	22.492	24.8931	76.9	0.027
PCL/CH/CMFE	19.5372	20.587	20.1107	69.2	0.030

## Data Availability

Not applicable.

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
