# Peer review of "Electrospun Polycaprolactone/Chitosan Nanofibers Containing *Cordia myxa* Fruit Extract as Potential Biocompatible Antibacterial Wound Dressings"

_molecules, 2023, doi:10.3390/molecules28062501_

Round 1

Reviewer 1 Report

The research subject of the reviewed work concerns the issues related to the fabrication of a wound dressing composed of electrospun nanofibers loaded with C. myxa fruit extract. The fibers were characterized in terms of their physical properties and antimicrobial effectiveness. Basically, the work is interesting however, I found in it a lot of important issues that aroused doubts in me during reading. You can find all my comments in the attached pdf file. I recommend a thorough improvement of the article.

Author Response

Thank you for your valuable comments concerning our manuscript entitled “Electrospun Polycaprolactone/Chitosan Nano-Fibers containing Cordia myxa Fruit Extract as a Potential Biocompatible Antibacterial Wound Dressings”. Those comments are all really valuable and helpful in revising and improving our manuscript, as well as providing significant direction for our research. We carefully considered the comments and made changes that we hope will be approved. The revised sections of the document are highlighted in red. The followings are the main corrections in the manuscript and responses to the reviewers’ comments.

Reviewer 2 Report

Title: Electrospun Polycaprolactone/Chitosan Nano-Fibers containing Cordia myxa Fruit Extract as a Potential Biocompatible Antibacterial Wound Dressings

Title: 

1. Authors have not conducted direct experiments related to wound dressings. just contemplated using the antimicrobial activity. it is suggested to change the title to antimicrobial activity.

Abstract:

1. Well written but can be improved with the incorporation of total results and conclusions.

Introduction:

1. Well written and suggested to include more latest references in place of old ones.

Methods:

1. All the methods used are highly standard and acceptable.

2. it is suggested to conduct direct wound healing experiments.

Results and discussions:

1. All the results well included in the manuscripts

2. All the results obtained were discussed in a good manner.

Conclusions:

1. rewrite this section with more appropriate information with clarity.

Reference:

1. Suggested reducing the number of references and including more latest references.

It is recommended to publish in the journal Molecules after correcting the above.

Author Response

Dear Editor and Reviewers,

Thank you for your letter and for the reviewers’ comments concerning our manuscript entitled “Electrospun Polycaprolactone/Chitosan Nano-Fibers containing Cordia myxa Fruit Extract as a Potential Biocompatible Antibacterial Wound Dressings”. Those comments are all really valuable and helpful in revising and improving our manuscript, as well as providing significant direction for our research. We carefully considered the comments and made changes that we hope will be approved. The revised sections of the document are highlighted in red. The followings are the main corrections in the manuscript and responses to the reviewers’ comments.

Reviewer 3 Report

Please consider the attached comments!

Good luck!

Author Response

(The authors gave the same response as above.)

Round 2

Reviewer 1 Report

Dear Authors

I recommend the article for publication but I still found some minor editorial errors in the work, e.g. I do not see a dot after the word Table 1, parenthesis should be removed in Table (2), and the word Inanofibrous shouldn't start with "I" letter.

Author Response

Reviewer #1:

 Dear Authors

I recommend the article for publication but I still found some minor editorial errors in the work, e.g. I do not see a dot after the word Table 1, parenthesis should be removed in Table (2), and the word Inanofibrous shouldn't start with "I" letter.

Author’s response: Author’s response: We are very grateful for the reviewers' insightful and valuable comments.

  1. 1. We have added a dot after the word in Table 1, and the parentheses have been removed in Table (2). 
    2. The word "Inanofibrous" has been corrected. (Please see the revised manuscript).

Reviewer 2 Report

Title: Electrospun Polycaprolactone/Chitosan Nano-Fibers containing Cordia myxa Fruit Extract as a Potential Biocompatible Antibacterial Wound Dressings

1. I think the old title is fine you can go with that only.

The Authors Have provided their responses to all the points raised and corrected and incorporated well in the revised Manuscript. Hence I recommend this MS for publishing in your reputed journal "Molecules" in the present form.

Author Response

Reviewer #2:

 Title: Electrospun Polycaprolactone/Chitosan Nano-Fibers containing Cordia myxa Fruit Extract as a Potential Biocompatible Antibacterial Wound Dressings

  1. I think the old title is fine you can go with that only.

The Authors Have provided their responses to all the points raised and corrected and incorporated well in the revised Manuscript. Hence I recommend this MS for publishing in your reputed journal "Molecules" in the present form.

Author’s response: We are very grateful for the reviewers' insightful and valuable comments.

The old title has been used.

Electrospun Polycaprolactone/Chitosan Nano-Fibers containing Cordia myxa Fruit Extract as a Potential Biocompatible Antibacterial Wound Dressings

(Please see the revised manuscript).
